# The Use of Nanomedicine to Target Signaling by the PAK Kinases for Disease Treatment

**DOI:** 10.3390/cells10123565

**Published:** 2021-12-17

**Authors:** Yiling Wang, Audrey Minden

**Affiliations:** Susan Lehman Cullman Laboratory for Cancer Research, Department of Chemical Biology, Ernest Mario School of Pharmacy, Rutgers, The State University of New Jersey, Piscataway, NJ 08854, USA; yw798@scarletmail.rutgers.edu

**Keywords:** nanomedicine, nanoparticles, PAKs, apoptosis

## Abstract

P21-activated kinases (PAKs) are serine/threonine kinases involved in the regulation of cell survival, proliferation, inhibition of apoptosis, and the regulation of cell morphology. Some members of the PAK family are highly expressed in several types of cancer, and they have also been implicated in several other medical disorders. They are thus considered to be good targets for treatment of cancer and other diseases. Although there are several inhibitors of the PAKs, the utility of some of these inhibitors is reduced for several reasons, including limited metabolic stability. One way to overcome this problem is the use of nanoparticles, which have the potential to increase drug delivery. The overall goals of this review are to describe the roles for PAK kinases in cell signaling and disease, and to describe how the use of nanomedicine is a promising new method for administering PAK inhibitors for the purpose of disease treatment and research. We discuss some of the basic mechanisms behind nanomedicine technology, and we then describe how these techniques are being used to package and deliver PAK inhibitors.

## 1. Introduction

Improper cell function can result in numerous disorders. For example, cancer can result from uncontrolled cell division, which can lead to the production of immature cells. These cells can spread to other parts of the body through the blood and lymphatic systems. Cancer-causing agents (carcinogens) can be present in the air, water, food, chemicals, and sunlight [1,2]. In addition to uncontrolled cell growth regulation and cancer, disrupted cell function can also cause other disorders, such as vascular disease, diabetes, and hemophilia. Although various treatments have been developed for disorders of cell growth and function, targeted therapy is especially promising. In targeted therapy, the specific inhibitors have been developed with the goal of obtaining a higher response rate, with more specificity and fewer side effects compared with more conventional treatments [3]. The PAK kinases have important roles in cell growth, survival, and migration, and are therefore often considered to be good targets for disease treatment. In addition to having important roles in normal cells, the PAKs are often improperly expressed in cancer and other diseases. In this review, we focus on the roles of the PAK kinases in disease, different inhibitors that have been generated against the PAKs, and the new use of nanomedicine to optimize delivery of PAK kinases.

### 1.1. The PAK Kinases

The PAKs are serine/threonine kinases that mediate multiple signaling pathways to control cellular functions including cell growth, cytoskeletal motility, cell proliferation, survival, and apoptosis [4,5]. The PAKs were originally identified as proteins that are activated by the Rho GTPases Rac1 and Cdc42, although they can also be activated by other mechanisms. The PAK family is classified into two groups, Group I consists of PAKs 1–3, and Group II consists of PAKs 4–6. These two groups of PAKs contain around 50% amino acid identity in the GTPase binding domain (GBD) and kinase domains, but outside of these regions they have sequence differences [6,7]. They can be distinguished by their tissue expression patterns in embryos, adults, and in diseases such as cancer. PAK4 is expressed in all tissues, but it is expressed at particularly high levels in embryogenic cells and some cancers. PAK1 is expressed at high levels in the brain, muscle, and spleen, while PAK2 expression is ubiquitous. PAK3, PAK5, and PAK6 tend to localize to embryonic and adult neuronal tissues [8].

### 1.2. PAK Kinases and the Regulation of Cell Survival and Apoptosis

The PAKs have several roles in the cell. Among these are important roles in cell survival and inhibition of apoptosis. Both PAK4 and PAK5, for example, have a protective effect against cell death. PAK4 induces cell survival by increased phosphorylation of the pro-apoptotic protein Bad (BCL2-associated agonist of cell death) and inhibition of caspase activation [9,10,11]. Similarly, PAK5 contributes to anti-apoptotic activity by phosphorylating BAD [12], or by activating Raf-1 via phosphorylation at serine 338, which targets Raf-1 to the mitochondria [13]. PAK4 can inhibit apoptosis by regulating caspase-3 cleavage [14]. PAK1 and PAK3 promote cell survival and inhibit apoptosis by phosphorylating BAD and Raf-1 [15,16]. PAK2 also has a unique role in cell apoptosis and survival. PAK2 can be activated both by Cdc42 and Rac, and by cleavage by caspase-3. Cleavage by caspase-3 can lead to apoptosis, but activation by Cdc42 or Rac can inhibit this cleavage, thus inhibiting apoptosis [17,18]. Overexpression of PAK1 decreases apoptosis and promotes cell growth signals via the AKT/mTOR signal pathway [19]. Recent studies show that PAK4 may also be associated with mTORC signaling, which in turn leads to regulation of AKT and cell survival [10]. The PAKs also stimulate other signaling proteins, such as JNK (c-Jun N-terminal kinase) and β-catenin, which is implicated in cell survival and protection from apoptosis [20]. Furthermore, suppression of PAK1 and PAK2 are distinctly associated with promoting cell apoptosis [21,22,23]. Thus, while PAKs have multiple different roles in the cell, regulation of cell survival and apoptosis is emerging as a key function of the PAKs, which is important for its role as a target for disease treatment. In addition to the regulation of cell survival and apoptosis, however, PAKs have other functions. Notably, the PAKs regulate cytoskeletal organization, and thus have crucial rules in cell migration, shape, and motility [9,24,25,26].

### 1.3. The PAKs as Potential Therapeutic Targets

Previous findings have shown that in some cases, increased cellular transformation, motility, survival, and tumor invasiveness are associated with an upregulation of specific PAKs, leading to cancer and metastasis [24,27]. Although the PAKs are not frequently mutated in human cancers, their overexpression can be associated with cancer [6].

While all the PAKs have important roles in cell growth and function, PAK1 and PAK4 are most often associated with tumorigenesis. Because of their overexpression in some types of cancer [5,28,29], several small molecule inhibitors have been developed to inhibit the PAKs [30], such as the Group I PAK inhibitors FRAX1036 [31], CP734 [32], FRAX597 [33], and IPA3 [4], and the Group II PAK inhibitors PF-3758309 [34,35], KPT-9274 [29,30,36], and GNE2861 [37]. Several of these inhibitors have shown promising results for inhibition of cell growth and transformation. IPA3 (“inhibitor targeting PAK1 activation-3”), for example, can induce cell death pathways in most leukemic cell lines and primary hematopoietic cells [38,39]. The PAK1 inhibitor CP734 suppresses pancreatic tumor growth in both in vitro and in vivo studies [32]. The miR-185 targets both PAK6 and BCR-ABL1. The restored expression of miR-185 remarkably represses the drug-resistant leukemic stem cells in chronic myeloid leukemia, and greatly improves the survival of leukemic mice [40]. The group I PAK inhibitors (G-5555 or FRAX1036), combined with the BRAF^V600E^ inhibitors (Vemurafenib) or AKT inhibitors (MK2206), effectively reduce cancer cell viability in a thyroid cancer mouse model in vivo [41], indicating the importance of combination therapies. KPT-9274 is a dual specific inhibitor of both PAK4 and NAMPT (nicotinamide phosphoribosyl transferase). It is effective at blocking cancer cell growth both in vitro and in vivo. For example, it can inhibit the growth of triple-negative breast cancer cells both in cell culture and in mouse models [10,42,43]. KPT-9274 is also growth inhibitory for several other in vivo cancer models, such as kidney cancer and pancreatic cancer [30,36,42,43]. Several clinical trials are still ongoing, to test the efficiency of KPT-9274 and other PAK inhibitors for the treatment of solid tumors, non-Hodgkins’s lymphoma, and myeloid leukemia (clinicaltrials.gov, last posted on 5 November 2021, 5 September 2021, 25 June 2021). PF-3758309, a small-molecule inhibitor targeting PAK4, can induce cell cycle arrest at the G1 phase, leading to apoptosis of neuroblastoma cells [34]. Suppression of PAK4 by PF-3758309 significantly prevents the growth of cisplatin resistant gastric cancer cells via the MEK/ERK and PI3K/Akt (PI3K: phosphoinositide 3-kinase) signal pathways. This is due in part to PAK4′s interaction with PI3K/Akt, and their mutual activation [35]. These results indicate that PAKs may be promising drug targets for cancer treatment, and that anti PAK drugs have the potential to be used either for the first line of defense or for adjuvant treatments for cancer.

The PAKs have multiple different roles in cell function, and thus their dysregulation has been associated with several disorders in addition to cancer. These include vascular diseases [44,45], neurodegenerative disorders [45,46], inflammatory diseases [47], allergic diseases [48], and asthmatic disease [49]. In vascular disease, the dysregulated growth of vascular smooth muscle cells (VSMC) is a critical factor in hypertension and atherosclerosis. PAK4 has been found to increase the proliferation of VSMCs in intimal hyperplasia (IH), which is an abnormal accumulation of vascular cells. This is mediated by AKT activation and downregulating p21, leading to cell cycle dysregulation. These results suggest a putative role for PAK4 as a therapeutic target for some types of vascular disease [44].

Neurodegenerative disorders, including Alzheimer’s disease (AD), Huntington’s disease (HD), and Parkinson’s disease (PD), are associated with certain cellular functions that can be regulated by the PAKs [50,51,52]. AD is the most common form of dementia for elderly people in the world. It is associated with deficits of dendritic spines and synapses. PAK1 and PAK3 have been shown to be aberrantly activated in AD, and subsequently translocate from cytosol to membrane, resulting in the loss of cytosolic PAK in the AD brain. The reduction of PAK in the cytosols of neuronal cells and leads to a decline in the actin binding protein Drebrin in dendritic spines, which is associated with memory loss in AD and other disorders [53,54]. This is consistent with the role for PAKs in regulating actin dynamics [55,56], and provides important insights into the contribution of PAKs to neurodegenerative diseases.

### 1.4. Nanoparticles/Nanomedicines

To further increase the efficacy of drugs, including those targeting PAKs, novel nanoparticles are currently being tested [38,57,58]. In recent years, nanoparticles (NPs) have attracted increasing attention as drug carriers, due to their excellent ability to stably encapsulate drugs, facilitate drug delivery, and prolong circulation time. Compared to free drugs, the use of nanoparticles could stabilize the pharmacokinetics of poorly water-soluble drugs, resulting in improved pharmacological activity, enhanced biodistribution [59,60,61], and even the ability to pass barriers such as the blood–brain barrier (BBB) [62]. Some of the early studies with NPs were carried out to deliver nutrition. For example, delivery of essential fatty acids and fat-soluble vitamins using NPs has been widely used as a significant source of energy and nutrition in the clinic [63]. This has had significant benefits for pediatric patients, critically ill patients, and long-term parenteral nutrition patients who had extensive intestinal resection [64,65,66].

More recently, NPs have been studied for their potential for disease treatment, including cancer. NPs fall into several different categories, including liposomes, exosomes, nanospheres, micelles, and nano-emulsions [67,68,69]. Among the different types of NPs, liposome and exosome-based NPs have been widely studied, and will be described in more detail here [69]. The liposome is a spherical vesicle with one or more phospholipid bilayers, ranging in diameter from 25 to 1000 nm. The phospholipid is an amphiphilic molecule composed of one hydrophilic head of phosphate and two hydrophobic fatty acid tails [70]. In aqueous solutions, the hydrophilic heads are inserted into water, and the hydrophobic tail extends into the air, self-spinning to form a spherical lipid bilayer (Figure 1A). The drug is encapsulated into either the aqueous core of the liposome or into the phospholipid bilayer, depending on whether the drug is hydrophobic or hydrophilic [70,71]. Liposomes are often considered to be efficient drug delivery systems. Their efficiency is due largely to biocompatibility, low toxicity, biodegradability, the ability to encapsulate hydrophilic or lipophilic anti-cancer drugs, and even the ability to target specific tumors or cancer cells [72].

Another major type of nanoparticle is the exosome. Exosomes (Figure 1B) are small extracellular vesicles (EV) secreted by mammalian cells, containing multiple proteins, lipids, and nucleic acids. Exosomes are attractive nanoparticles due to their stability in circulation, efficiency in cellular uptake, low immunogenicity, and low toxicity [73]. Exosomes are produced in the endosomal system, by the small intraluminal vesicles (early exosomes) in the multivesicular bodies (MVBs). When the MVB fuses with the plasma membrane, the mature exosomes are released to the extracellular environment [74]. Exosomes are also called natural biomimetic nanoparticles, because they combine specific functions of cells or cell membranes, and they have the multifunctionality of synthetic nanoparticles [69]. The aqueous cores of exosomes can encapsulate siRNAs or hydrophilic drugs and protect them from nucleases, while hydrophobic drugs can be encapsulated by their membranes. Nevertheless, exosomes have some limitations compared with synthetic NPs, including low volume production, cargo loading capacity, and quality control of exosomes [75]. To overcome these limitations, exosomes have been developed for loading special DNAs, peptides, or proteins by expressing specific DNA-binding domains and antibodies [75]. Compared to other NPs, these modified exosomes display lower immunogenicity, longer half-life, and better biostability and biocompatibility [75].

In some studies, the different types of NPs are being designed based on enhanced permeability and retention (EPR) effects. This is especially important for the treatment of solid tumors since their anatomical and pathophysiological characteristics are different from normal tissue. EPR refers to the unique structures of vessels at the sites of solid tumors. These vessels are usually leaky, with large gaps between the endothelial cells, and they show selective extravasation and retention of macromolecules [76]. This has led to the design of NPs that can specifically cross these barriers in the vessels. NPs have been generated that accumulate more readily within tumors, compared with the normal tissues [76]. Although small nanocarriers of approximately 400 nm can be delivered into tumors due to the EPR effect, some pathophysiological characteristics of tumors still make drug delivery difficult, such as abnormal vasculature, pH, temperature, and surface charge surrounding the tumor cells [77]. To overcome these problems, specific NPs have been developed, such as pH-sensitive liposomes [78], temperature-sensitive liposomes [79], and polymeric nanoparticles with reversal surface charge at the tumor site [80].

Although EPR is groundbreaking for treating human tumors, a relatively small number of treatments utilizing this technology have become clinical options for human patients [76,81]. Furthermore, optimizing the conditions in human patients is limited, due to ethical concerns. Ding et al. [81] have developed a new strategy for investigating NPs and EPR effect in human tumors. They found that the EPR effect in humans correlates with certain factors such as tumor size and patients’ gender. Their work indicates that NPs and EPR effects remain promising, and that conditions can still be optimized to use them most effectively.

Although NPs are good carriers for drugs, especially poorly water-soluble drugs, they do have some potentially limiting factors that must be overcome. NPs have some safety concerns, including the possibility for off-targeted toxicity, and other potential therapeutic risks [82]. Despite these limitations, the use of NPs is promising for the treatment of cancer and other diseases. Here, we describe several examples of recent studies in which NPs are used to encapsulate the PAK inhibitors for use in research and disease treatment.

### 1.5. Use of Nanomedicine to Target PAK Kinases

Improper expression of the PAK kinases has been implicated in several disorders, but their overexpression in cancer is particularly notable [5,26,83]. While surgery and chemotherapy are traditional approaches for cancer treatment, these approaches are not always effective, and many chemotherapeutic agents have non-specific side effects [72,77,80]. The use of more specific inhibitors is therefore promising. Among these are inhibitors of the PAKs, for which there have been some promising results [41,84]. However, even when using specific inhibitors such as PAK inhibitors, poor pharmacokinetics profiles are often a problem [26,85]. NPs have attracted attention as drug carriers, due to their excellent ability to stably encapsulate drugs, facilitate drug delivery, and prolong circulation time [86]. As discussed throughout this proposal, several studies have shown more promising results when PAK inhibitors are combined with the use of nanomedicine, compared with free PAK inhibitors [38,57,84,86]. The use of nanoparticles can stabilize the pharmacokinetics of poorly water-soluble drugs, resulting in improved pharmacological activity, enhanced biodistribution, and even the ability to pass barriers such as the blood–brain barrier (BBB) [87,88]. As described above, NPs are also being designed to specifically target tumors [77]. This review, therefore, will focus on the use of PAKs combined with nanoparticle technology, as a promising emerging strategy for the treatment of cancer and other diseases.

## 2. Inhibition of Pak1 for the Treatment of Prostate Cancer

PAK1 is known to be expressed at high levels in prostate cancer, although its expression is relatively low in normal prostate. PAK1 has in fact been shown to be essential for the growth of prostate cancer cells in a xenograft mouse model [86,89]. PAK1 contributes to prostate cancer cell growth and metastasis, due to its role in cell proliferation, survival, motility, invasion, and epithelial–mesenchymal transition (EMT) [90,91,92,93]. Inhibition of PAK1 is therefore promising for the treatment of this disease. Several studies discussed below, including in vivo studies, have shown that inhibition of PAK1 can reduce prostate cancer cell growth. IPA-3 is a PAK1 inhibitor that has been considered promising for the possible treatment of prostate cancer [86,91]. A drawback to this inhibitor, however, is that it is metabolically unstable. Al-Azayzih et al. [86] have found a way to circumvent this problem, by encapsulating the inhibitor in sterically stabilized liposomes (SSL), which are lipid-based nanoparticulate drug carriers. The particles average 139 nm in diameter. This method can facilitate drug delivery and decrease off-target toxicity. This new drug (SSL-IPA3) had an increased half-life compared with free IPA-3. SSL-IPA3 was tested for its effects on prostate cancer cell viability in vitro, and for growth of PC-3 prostate cell tumors in an in vivo mouse model. Both SSL- and SPRL-IPA3 have markedly anti-proliferative effects on murine prostate cells (RM-1 cells). However, the free IPA3 had no significant therapeutic efficacy in the transgenic adenocarcinoma of the mouse prostate mice [90]. The authors found that this nanoparticle delivery system resulted in strong efficiency both in vitro and in vivo in the prostate cancer model, and that the drug showed strong specificity.

A second method for delivery of IPA3 to prostate cancer cells, also utilizing liposome delivery, involved the targeting by secreted phospholipase A2 (sPLA2), which is an esterase that is overexpressed in prostate cancer and several other cancers [90]. The sPLA2 responsive liposomes (SPRL) are based on SSLs, but are altered to include an increase in glycophosplipids, which have a negative charge. This makes these complexes more responsive to cancers that overexpress SPLA2, including prostate and other cancers [38,90,94]. Prostate cancer cells often have high levels of sPLA2 on their surface [95], and SPRLs have been shown to have a high affinity to these cell surface sPLA2 moieties. Verma et al. [90] tested the use of both SSL-IPA3 and SPRL-IPA3 using in vitro and in vivo models of colon cancer, and also investigated lung metastasis. Both mechanisms showed increased efficiency, compared with free IPA3 alone. SPRL-IPA3 had an even higher efficacy in reducing cell survival than SSL-IPA3, but both were similar in efficiency and significantly more effective than free IPA3. In both cases, the treatment with the liposome encapsulated forms of IPA3 could be carried out significantly less frequently than free IPA3 in mouse models. Importantly, metastasis to the lungs was also significantly reduced. These results suggest that nanomedicine delivery of the PAK1 inhibitor could be a promising mechanism for the treatment of prostate and other cancers.

## 3. Inhibition of Pak1 for the Treatment of Triple Negative Breast Cancer

In addition to prostate cancer, PAK1 is often upregulated in other cancers, including breast cancers. The SPRL-IPA3 and SSL-IPA3 complexes described above have therefore also been tested for growth inhibition of breast cancer cells [38]. A panel of breast cancer cells representing different stages and subtypes of the disease were analyzed. While both compounds were effective at growth inhibition, SPRL-IPA3 appeared to be particularly effective for inhibiting the growth of metastatic Triple Negative Breast Cancer (TNBC) cells and functioned by causing apoptosis in these cells. This is consistent with previous findings that IPA-3 can lead to increased expression of apoptotic proteins such as caspase-3 and caspase-9 [38,39,92]. Interestingly, sPLA2 may be a prognostic factor for disease recurrence in breast cancer [38,96]. This is consistent with the finding that SPRL-IPA3 was more effective than SSL-IPA3 in inhibiting the growth of breast cancer cells [38]. These studies suggest that nanomedicine delivery of the PAK1 inhibitor may be effective for breast cancer as well as prostate cancer and warrants further investigation.

## 4. Inhibition of PAK4 Prolongs Survival in a Pancreatic Cancer Mouse Model

Pancreatic cancer has a particularly low prognosis, due to difficulty in both diagnosis and treatment. Pancreatic cancer is often associated with increased levels of Pak4 [29,57,97]. This is consistent with a role for PAK4 in cell survival, anchorage independent growth, and migration [5,57,97]. In vitro studies have shown that inhibition of PAK4 with siRNA and other mechanisms, can reduce the growth of pancreatic cancer cells [57]. However, until recently, few studies have validated its role as a target for pancreatic cancer in vivo. RNA interference (RNAi) is a promising mechanism for knocking down target genes. However, its promise in cancer treatment is limited, because of delivery problems, as well as toxicity, and because the molecules are somewhat large. A new potential mechanism for delivery of RNAi is via exosomes. Exosomes, as described above, are naturally occurring nanoparticles in the form of extracellular vesicles, of about 30–150 nm in size [69]. siRNA can be encapsulated in the aqueous cores of exosomes, and thus they will be protected from nucleases. Packaging in the exosome will also facilitate cellular uptake [98]. PAK4 siRNA was loaded into exosomes to form Exo-siPAK4 and purified. Exo-siPAK4 and free siPAK4 were compared, by transfection into the pancreatic cancer cells PANC-1, using electroporation. Exo-siPAK4 was efficiently transfected into the cells in vitro, and PAK4 expression was inhibited. Even more importantly, in vivo results were also promising. Intratumoral injection of the Exo-siPAK4 in resulted in effective uptake into the tumors, and PAK4 was efficiently knocked down. Tumor growth was inhibited, and survival times of the animals increased. These effects were temporary, however, suggesting that further doses would be warranted. Although the dosage and timing may need to be optimized, this study indicates that exosome encapsulated siRNA may be an effective way to treat pancreatic and other cancers.

## 5. Cocktail Therapy (Combinational Treatment) Delivered by Nanoparticles to Treat Hepatocellular Carcinoma (HCC)

Hepatocellular carcinoma (HCC), a major type of cancer of the liver, still lacks efficient treatment. Multiple different genes have been shown to be associated with HCC. Among these are PAK4. PAK4 is highly expressed in HCC, and its inhibition can reduce proliferation of HCC cells [84,99,100]. MicroRNAs (miRNAs) have also been shown to have important roles in HCC as well as other cancers. miRNAs are endogenous noncoding RNAs that regulate genes by binding to the 3’—untranslated regions (UTRs) of mRNA [84,100,101,102]. miRNAs can be improperly expressed in some types of cancers and can act as either tumor suppressors or promoters [103]. miRNA-433 is known to function as a tumor suppressor. Its levels are reduced in HCC, and this correlates with high levels of PAK4. miRNA-433 is thus thought to inhibit HCC cell proliferation in normal cells by targeting PAK4. Another important miRNA is miR-199a-3p, which is highly expressed in normal liver but downregulated in nearly all HCCs. Downregulation of miR-199a-3p is associated with poor prognosis [99]. This miRNA targets several genes including PAK4 [99,104,105] and MTOR [106], and subsequently leads to the repression of oncogenic FOXM1 transcription factor [99,107]. miR-199a-3p has shown promise in in vivo studies. In vivo delivery of miR-199a-3p has strong anti-tumor activity, making it a promising therapeutic option [99,104,106]. Conversely, miR-10b, can act as an oncogene and increase metastasis in HCC [108,109]. Regulating the levels of these and other miRNAs could therefore have profound effects. Because of the role for miRNAs in regulating cancer cell growth, there has been a great deal of interest in miRNA-based therapeutics. While this idea is promising, optimal mechanisms for miRNA use and delivery still need to be developed. For example, the use of only a single miRNA is not always sufficient [110], and HCC is a heterogeneous disease caused by the dysregulation of multiple genes. To maximize the efficiency of miRNAs, the cocktail strategy has been employed, where two or more miRNAs are combined [84,111]. Shao et al. [84] describe the use of a customized cocktail of miRNA therapeutics for the treatment of HCC. This treatment involves the nanosystem PCACP (PEI-betaCD@Ad-CDM-PEG) for targeting delivery to HCC. The PCACP is a polymer-based nanoplatform for miRNA therapy, which is cleaved in response to mild acidity at the tumor site. This delivery strategy allows highly efficient transfection, high tumor uptake, and precise control of delivery. Shao et al. [84] used this strategy for encapsulating miR-199a/b-3p mimics along with anti-miR-10b to prevent the growth of HCC. The miR-cocktail strategy was more efficient compared with treatment by miR199 or antimiR10b alone and led to reduction in HCC both in vitro and in vivo.

The study described above depends in part on the use of the miR-199a/b-3p. This miRNA has been shown to target PAK4 and mTOR, and to suppress EMT. In contrast, miR10b is an oncogene, and also a driver, which can induce metastasis and facilitate the EMT process in HCC via the HOXD10, CAMD, and RHOC pathways [84]. Anti-miR10b is thus considered an adjuvant treatment to be used with miR-199a/b-3p to enhance its anti-metastasis effect in HCC. The data showed that the PCACP/miR-cocktail leads to significant downregulation of the PAK4 and mTOR pathways, leading to suppressed HCC pathogenesis. This study indicates that the PCACP/miR-cocktail nanosystem notably inhibited tumor progression and multitarget regulated by mTOR, PAK4, RHOC, and EMT pathways, which resulted in improved therapeutic effect for HCC treatment. This is a promising strategy for future personalized treatments of HCC or other types of cancer.

## 6. Inhibition of Pak4 for the Treatment of Atherosclerosis

Atherosclerosis is a leading cause of cardiovascular disease. It is characterized initially by deposits of lipoprotein in the inner walls of the arteries. The disease is associated with chronic inflammation, as well as the accumulation of Low-Density Lipoprotein (LDL) particles. LDL particles become oxidized by Reactive Oxygen Species (ROS) or by products of macrophages. During atherosclerosis, monocytes invade the endothelial tissue of the arteries, and inflammatory macrophages develop from these monocytes. Oxidized LDL (oxLDL) can facilitate incorporation of cholesterol into macrophages via the CD36 receptor. The macrophages then eventually develop into “foam cells”, key components of atherosclerosis, which ingest LDL particles. This in turn leads to expression of cytokines such as Interleukin-6 (IL-6) and Macrophage Chemotactic Protein (MCP), resulting in plaque formation and smooth muscle migration. Overexpression of the cell surface protein CD36 on macrophages is a key feature of atherosclerosis, leading to uptake of oxLDL. Blocking the oxLDL/CD36 interaction, therefore, may be an important mechanism for disease treatment. Studies have suggested that blocking PAK1 may suppress internalization of oxLDL as well as other factors such as IL-6 and MCP.

A promising mechanism for blocking PAK1 is the use of small interfering RNA (siRNA). Wu et al. [112] tested the use of siRNA nanomedicine treatment for atherosclerosis. This involved the use of CD36 antibody-directed anti PAK1 siRNA nanomedicine. PAK1 siRNA was packaged by complexation with poly aspartic acid, grafted with low molecular weight poly-ethyleneimine. The goal was to suppress the expression of proinflammatory factors such as MCP-1 and IL-6, which may be regulated at least in part by PAK1. Since the complex also contained CD36 antibody, another goal was to downregulate the CD36 receptor, thus further reducing atherosclerosis. This complex was tested both in vitro, using RAW 264.7 macrophage cells, and in vivo, in a mouse model. Uptake of the complex was efficient, and inflammatory factors (MCP-1 and IL-6) were suppressed in response to PAK1 silencing. In vivo, the complex accumulated at atherosclerotic plaques and was taken up by macrophages to silence PAK1. This led to reduction in IL-6 and MCP-1 expression, and decreased oxLDL deposition in macrophages. As a result, atherosclerosis was significantly reduced in this in vivo model systems. These results demonstrate an important role for PAK1 in the regulation of signaling pathways involved in the induction of inflammatory cytokines. They also illustrate the promising possibility that targeting PAK1 with nanomedicine siRNA is a promising treatment for atherosclerosis and possibly other diseases involving inflammation.

## 7. Conclusions and Future Perspectives

In this review, we describe multifunctional NP formulations (including liposomes and exosomes) and discuss that they can be used to deliver PAK inhibitors. The use of NPs is an important new mechanism for the development of novel drugs to provide better delivery, overcome drug resistance, and to directly target tissues and cells. We also discuss that there is an important role of PAKs in various diseases, including cancers, atherosclerosis, and some neurological disorders. We describe that targeting PAKs combined with nanoparticle technology is a promising emerging strategy for the treatment of cancer and other diseases involving PAK kinases.

The use of nanomedicine for delivery of drugs such as PAK inhibitors is quite encouraging. Nevertheless, since single agent treatments are not always sufficient, in the future, the combination of different types of treatments and the targeting of multiple pathways may be key for disease treatment. Nanoparticle systems are particularly promising, for reasons discussed in this review. Nevertheless, limitations remain, and modified nanoparticle formulations are still in development. Furthermore, because of the heterogeneity of diseases and individuals, personalized approaches to treatment in different individuals is becoming increasingly important. Overall, new approaches to disease treatment that include the use of nanomedicine and inhibition of kinases such as PAK kinases constitute a promising new direction for disease treatment.

## Figures and Tables

**Figure 1 cells-10-03565-f001:**
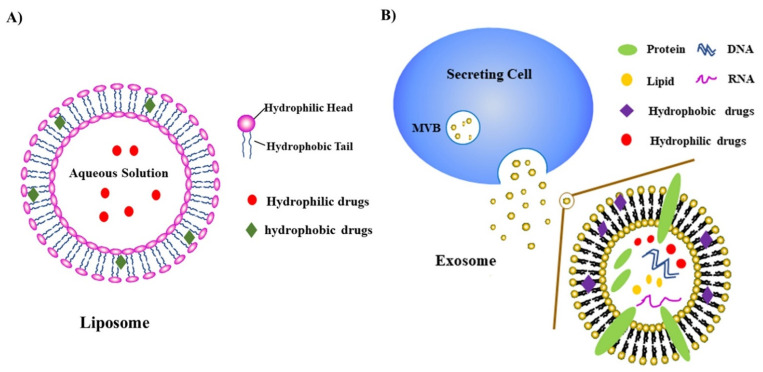
Structure of liposomes (**A**) and exosomes (**B**). (**A**) The liposome is a spherical vesicle with one or more phospholipid bilayers. Drugs can be encapsulated in the aqueous core of the liposome or within the phospholipid bilayer. (**B**) Exosomes, which can also encapsulate various drugs and particles, are small extracellular vesicles (EVs) secreted by mammalian cells, containing multiple proteins, lipids, and nucleic acids. They are also referred to as natural biomimetic nanoparticles.

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
