# Peer review of "The Use of Nanomedicine to Target Signaling by the PAK Kinases for Disease Treatment"

_cells, 2021, doi:10.3390/cells10123565_

Round 1
Reviewer 1 Report
This review by Yiling Wang and Audrey Minden discussing the targeting of P21 activated kinases (PAKs) with nanomedicine for disease treatment. The topic of this review seems to be interesting but not convincing and at certain degree confusing. Please find below some general and specific comments/questions: What is the goal of this review? It sems to be too broad and might confuse readers as it highlights too many different aspects. For instance, authors discussing family of PAKs and different therapeutic approaches which would target different members of PAK family. Next level of complexity here - are different diseases such as Alzheimer, Parkinson’s, Huntington’s disease, breast cancer, prostate cancer, hepatocellular carcinoma, and the use of different nanomedicine approaches in those by targeting variety of PAK’s delivered by exosomes, sterically stabilized liposomes, polymer based nanoplatforms etc. Altogether, it is potentially confusing for the readers who are not the experts in this fields. Specific questions/comments: 1. Could you provide evidence why targeting PAKs is still relevant for clinic - (data from clinical trials, more in vivo works, rather than papers based on the data from in vitro only)? 2. Could you justify why using nanoparticles to target PAK’s is more effective than other approaches and classify it in the context of different diseases. 3. Would it be possible to discuss diseases where PAK’s targeting by nanomedicine is not effective and why? 4. It is not clear, what is a correlation between type of disease and type of PAK’s targeted nanomedicine? 5. Lack of discussion with the future perspectives for nanomedicine in targeting PAK’s. It seems to be that readers will need to draw their own conclusions about discussed topic without any analysis from authors what makes this work just more descriptive then analytic. Minor remarks: 1. 39 “In targeted therapy inhibitors are developed against specific targets” – should be rephrased. 2. 147 “Liposomes are often considered to be an efficient drug delivery system for drugs.” - “Drug delivery system” – this term speaks by itself about what it designed for.Author Response
Dear Editor,
Please see the reply below and the revised review in the attachment.
Thank you very much for the positive review of our review article entitled “The use of nanomedicine to target signaling by the PAK kinases for disease treatment”, by Wang and Minden, for publication in Cells. We have responded to the reviewer's critiques, and have included a point-by-point response to their concerns. The reviewer's comments were helpful, and we believe the manuscript is even stronger now due to their suggestions.
Thank you very much for your consideration.
Sincerely,
Audrey Minden, Ph.D.
Associate Professor
Response to Reviewer 1 Comments:
Point 1: The reviewer has pointed out that our paper would benefit from more focus, and emphasis on the goal of the review. Our response is below, and we have made the corresponding changes to the text as indicated.
Response 1: The goal of this review is to describe new strategies involving the use of using nanomedicine encapsulated drugs for targeting PAK Kinases in disease treatment. We indicate that this could be a promising approach, and that the combination of targeting PAKs and using nanomedicine techniques may overcome some of the limitations of conventional drug systems. This is due to several factors including improvement in biodistribution, and avoidance of adverse side effects. Many previous studies focused on either the PAKs, or nanomedicine alone. The use of combination therapy, in which NPs are used to target the PAKs, is relatively new. Our review is comprehensive, but also focused. The focus is on three major areas: (a) Targeting the PAK kinases, (b), the basic methodology of nanomedicine, and (c) How nanomedicine may be used in combination with PAK inhibitors. The overall focus is highlighted more clearly in the revised manuscript, in the abstract, and in the introduction. Although cancer is the disease that we give the most attention to, we felt it was important to briefly discuss the other diseases that PAK kinases may be involved in as well.
Point 2: The reviewer asks for evidence as to PAKs relevancy in the clinic.
Response 2: We discuss several examples of PAK inhibitors being used for in vivo studies. KPT-9274, a dual specific inhibitor of PAK4 and NAMPT, has been tested in vivo in several mouse cancer models including TNBC, and has been shown to reduce tumor growth. Several clinical trials are also now ongoing to test the efficiency of this PAK4 inhibitor in cancer. In addition, several recent studies have shown that inhibition of PAK1, PAK4, and PAK6, reduce tumorigenesis in vivo. These studies are highlighted more clearly in the revised manuscript, in section 1.1, and elsewhere throughout the paper. In many examples, results were more promising when nanomedicine was used, rather than traditional methods, which is a key point of this review. For this reason, although clinical trial data is limited at this time, there is reason to believe that with the use of new delivery technology, future results will be quite promising.
Point 3: The reviewer asks that we justify why using nanoparticles to target PAK’s is more effective than other approaches.
Response 3: While the purpose of this review is not to indicate that using NPs is always more effective than other methods, the use of NPs is a promising new technology. A major emphasis of this review article is cancer, although we also discuss the roles for PAKs in other diseases. While surgery and chemotherapy are traditional approaches for cancer treatment, these approaches are not always effective, and many chemotherapeutic agents have non-specific side effects. The use of more specific inhibitors is therefore promising, but even in this case, poor pharmacokinetics profiles are often a problem. NPs have attracted attention as drug carriers, due to their excellent ability to stably encapsulate drugs, facilitate drug delivery, and prolong circulation time. Compared to free drugs, the use of nanoparticles can stabilize the pharmacokinetics of poorly water-soluble drugs, resulting in improved pharmacological activity, enhanced biodistribution, and even the ability to pass barriers such as the blood-brain barrier (BBB). A major barrier for treating solid tumors is delivering drugs directly to the tumor with maximum efficacy, since their anatomical and pathophysiological characteristics are different from normal tissue. For this reason, some NPs are being designed based on enhanced permeability and retention (EPR) effects. This has led to the design of NPs that can specifically cross barriers in the vessels. NPs have been generated that accumulate more readily within tumors, compared with the normal tissues. Since PAKs have important implications in cancer and other diseases, targeting PAKs combined with nanoparticle technology is a promising emerging strategy for the treatment of cancer and other diseases involving PAK kinases. These advantages to using nanoparticles are highlighted in the revised paper, in a new section, 1.3.
Point 4: The reviewer asks for a discussion of diseases where PAK’s targeting by nanomedicine is not effective.
Response 4: Currently using nanomedicine to target PAKs is relatively new. Hence there have not been a great deal of negative results that have been published. There are numerous examples, however, of cancers and other diseases that are not associated with PAK overexpression, and it is not expected that targeting PAKs with nanomedicine would work for every situation. Here we focus, however, on several examples of diseases where PAK expression has been implicated, and where encouraging results have been obtained.
Point 5: The reviewer asks about the correlation between the type of disease and the type of PAK’s targeted nanomedicine?
Response 5: The focus of the review is primarily on PAKs and cancer although we felt it was important to mention other diseases for which PAK kinases are also implicated. In terms of cancer, we mention in the text that the PAKs that have most often been implicated in cancer are PAK1 and PAK4. This is indicated in section 1.1 in the revised proposal. We have also made sure that we mention which PAK family member is involved in the different studies we discuss throughout the manuscript.
Point 6: This question is in regards to a lack of discussion with the future perspectives for nanomedicine in targeting PAKs.
Response 6: We have added a paragraph at the end of the manuscript, with more information about conclusions and future perspectives.
Point 7: Minor remark 1: 39 “In targeted therapy inhibitors are developed against specific targets” – should be rephrased.
Response 7: We have revised the sentence pointed out by the reviewer, which now reads “In targeted therapy, specific inhibitors have been developed”.
Point 8: Minor remark 2: 147 “Liposomes are often considered to be an efficient drug delivery system for drugs.” - “Drug delivery system” – this term speaks by itself about what it designed for.
Response 8: We revised the sentence pointed out by the reviewer, which now reads: “Liposomes are often considered to be efficient drug delivery systems”.

Reviewer 2 Report
The manuscript gives ad over view about the PAKs and how can be administered using nanotechnology. It also gives a review of the nanoparticulate system being developed using this approach. However, there are some issues that ca be resolved to make the manuscript worthy.
- There are many typographical errors. E.g., line 60 (pro-apoptotic Bad protein) or is it BAD. it should be consistent through, either in lowercase or uppercase.
- Few other typing errors: line 61, 64, 113, 208
- Overall, the manuscript should be cross checked for the grammar. E.g., line, 71, 75, 136.
- It would be really nice if the ongoing clinical trials can be mentioned in few lines.
Author Response
Dear Editor,
Please see the reply below and the revised review in the attachment.
Thank you very much for the positive review of our review article entitled “The use of nanomedicine to target signaling by the PAK kinases for disease treatment”, by Wang and Minden, for publication in Cells. We have responded to the reviewer's critiques, and have included a point-by-point response to their concerns. The reviewer's comments were helpful, and we believe the manuscript is even stronger now due to their suggestions.
Thank you very much for your consideration.
Sincerely,
Audrey Minden, Ph.D.
Associate Professor
Response to Reviewer 2 Comments:
Point 1and 2: Questions 1 and 2 regard typographical errors and our use of the term BAD protein.
Response 1 and 2: Typographical errors throughout the text have now been fixed. Bad has been revised and is now BAD. We also indicate that it stands for BCL2-associated agonist of cell death (a pro-apoptotic protein).
Point 3: Overall, the manuscript should be cross checked for the grammar. E.g., line, 71, 75, 136.
Response 3: As requested by the reviewer, we have fixed all grammatical and typographical mistakes.
Point 4: It would be really nice if the ongoing clinical trials can be mentioned in few lines.
Response 4: As the reviewer suggested, we mentioned some of the ongoing clinical trials, and have emphasized in vivo studies that involve PAK inhibitors. This can be seen in the revised section 1.1.

Round 2
Reviewer 1 Report
The authors have responded to all my questions and comments therefore I have no further concerns that require another revision.